# Association of physical activity and sleep habits during pregnancy with autistic spectrum disorder in 3-year-old infants

Kazushige Nakahara [1,26], Takehiro Michikawa[2,26], Seiichi Morokuma [3,4✉], Norio Hamada [1,4], Masanobu Ogawa[1,4], Kiyoko Kato[1,4], Masafumi Sanefuji[4,5], Eiji Shibata[6,7], Mayumi Tsuji[6,8], Masayuki Shimono[6,9], Toshihiro Kawamoto[6], Shouichi Ohga[4,5], Koichi Kusuhara[6,9] & the Japan Environment and Children's Study Group*

## Abstract

**Background:** We hypothesized that maternal lifestyle factors, such as physical activity and sleep habits, may be associated with autism spectrum disorder (ASD) in infants. This study aimed to investigate the association between maternal physical activity and sleep before and during pregnancy with infant ASD diagnosed by the age of 3 years.

**Methods:** We used the data from the Japan Environment and Children's Study between 2011 and 2014. The study included 103,060 pregnant women, among which, 69,969 women were analyzed. Participants were asked about their physical activity and sleep before and during pregnancy using questionnaires during pregnancy. Maternal physical activity was estimated using the international physical activity questionnaire. Based on the levels of physical activity before or during pregnancy, the participants were divided into five groups. Maternal sleep was analyzed based on sleep duration and bedtime. The outcome was diagnosis of ASD in 3-year-old infants.

**Results:** In mothers with higher physical activity levels during pregnancy, the risk ratios (RR) for ASD in their 3-year-old infants were lower (RR = 0.61, 95% confidence interval (CI) = 0.42–0.90). In contrast, too short (<6 h) and too long (>10 h) sleep durations during pregnancy were associated with higher risk ratios for ASD than 7–8 h sleep duration (too short: RR = 1.87, 95% CI = 1.21–2.90; too long: RR = 1.56, 95% CI = 1.00–2.48). These associations were not observed before pregnancy.

**Conclusion:** Maternal physical activity and sleep duration during pregnancy may be associated with ASD in infants.

## Plain language summary

Children with autism spectrum disorder (ASD) have difficulties with communication and can have problematic behavior. It is unclear whether lifestyle habits before and during pregnancy influence the chances of having a child with ASD. We investigated whether maternal physical activity and particular sleep habits before and during pregnancy increased the likelihood of 3-year-old infants having ASD. Mothers completed questionnaires asking about their physical activity, sleep habits, and any diagnosis of ASD in their child. The children of mothers with high levels of physical activity during pregnancy were less likely to have ASD. The children of mothers who had slept for less than 6 h or longer than 10 h a night were more likely to have ASD. Thus, improving sleep and increasing exercise during pregnancy might reduce the risk of ASD in their children.

[1] Department of Obstetrics and Gynecology, Graduate School of Medical Sciences, Kyushu University, Fukuoka, Japan. [2] Department of Environmental and Occupational Health, School of Medicine, Toho University, Tokyo, Japan. [3] Department of Health Sciences, Graduate School of Medical Sciences, Kyushu University, Fukuoka, Japan. [4] Research Center for Environment and Developmental Medical Sciences, Kyushu University, Fukuoka, Japan. [5] Department of Pediatrics, Graduate School of Medical Sciences, Kyushu University, Fukuoka, Japan. [6] Regional Center for Japan Environment and Children's Study (JECS), University of Occupational and Environmental Health, Kitakyushu, Japan. [7] Department of Obstetrics and Gynecology, School of Medicine, University of Occupational and Environmental Health, Kitakyushu, Fukuoka, Japan. [8] Department of Environmental Health, School of Medicine, University of Occupational and Environmental Health, Kitakyushu, Fukuoka, Japan. [9] Department of Pediatrics, School of Medicine, University of Occupational and Environmental Health, Kitakyushu, Fukuoka, Japan. [26]These authors contributed equally: Kazushige Nakahara, Takehiro Michikawa. *A list of authors and their affiliations appears at the end of the paper. ✉email: morokuma.seiichi.845@m.kyushu-u.ac.jp

The prevalence of developmental disorders such as autism spectrum disorder (ASD) is increasing in developed countries, including Japan[1–3]. ASD is associated with various prenatal, perinatal, and postnatal risk factors, including genetic and environmental factors[4–6].

Maternal lifestyle appears to be associated with offsprings' development. For example, higher maternal physical activity (PA) during pregnancy has been reported to be positively associated with infants' development. Two review articles in 2018 concluded that PA during pregnancy was associated with improved language development[7] and total neurodevelopment of the offspring[8]. Another example of such a maternal lifestyle factor is maternal sleep. Short duration of maternal sleep or late bedtime during pregnancy is associated with preterm birth and gestational diabetes mellitus (GDM)[9,10], which are also reported risk factors for ASD[11].

Maternal inflammation and metabolic disorders are possible causal pathways that link maternal lifestyle with infant ASD. Maternal inflammation has been shown to be associated with maternal PA and sleep[12–15], and uterine inflammation is considered to be a cause of developmental disorders[16,17]. Similarly, maternal metabolic disorders, such as obesity and dyslipidemia, affect maternal PA and sleep[15,18], and may be associated with infant developmental disorders through hypertensive disorders in pregnancy (HDP) and GDM[19–23].

For the above reasons, we hypothesized that maternal lifestyle factors, such as PA and sleep habits, may be associated with ASD in their children. However, to the best of our knowledge, these associations have not been reported in interventional studies, such as randomized controlled trials, or in observational studies. In addition, the influence of preconception maternal lifestyle on offspring development has not been reported. We previously reported that higher levels of maternal PA both before and during pregnancy are associated with a decrease in sleeping and developmental problems in 1-year-old infants[24]. Moreover, maternal sleep habits, such as short sleep duration and late bedtime, both before and during pregnancy, are associated with an increase in sleeping problems in 1-year-old infants[25]. Children with ASD frequently have sleeping problems, such as late bedtime and intense night crying, and developmental abnormalities from early infancy[26–28]. We hypothesized that maternal PA and sleep habits before and during pregnancy are associated not only with early infancy sleep and developmental problems but also with the subsequent diagnosis of ASD.

In this observational study, we aimed to investigate the association between maternal PA and sleep habits before and during pregnancy and ASD diagnosis in their 3-year-old toddlers. The result of this study showed that higher maternal PA levels during pregnancy may be associated with a lower risk of ASD in 3-year-old infants, whereas too short or too long sleep durations during pregnancy may be associated with a higher risk of ASD in 3-year-old infants. Our findings suggested that maternal lifestyle during pregnancy may be associated with risk of ASD in their children.

## Methods

**Research ethics**. The Japan Environment and Children's Study (JECS) protocol was reviewed and approved by the Ministry of the Environment's Institutional Review Board on Epidemiological Studies (no. 100910001) and by the Ethics Committees of all participating institutions: the National Institute for Environmental Studies that leads the JECS, the National Center for Child Health and Development, Hokkaido University, Sapporo Medical University, Asahikawa Medical College, Japanese Red Cross Hokkaido College of Nursing, Tohoku University, Fukushima Medical University, Chiba University, Yokohama City University, University of Yamanashi, Shinshu University, University of Toyama, Nagoya City University, Kyoto University, Doshisha University, Osaka University, Osaka Medical Center and Research Institute for Maternal and Child Health, Hyogo College of Medicine, Tottori University, Kochi University, University of Occupational and Environmental Health, Kyushu University, Kumamoto University, University of Miyazaki, and University of Ryukyu. This study was conducted in accordance with the Declaration of Helsinki. Written informed consent, which was also obtained for a follow-up study of the children after birth, was obtained from all recruited pregnant women.

**Study participants**. Data used in this study were obtained from the JECS, an ongoing large-scale cohort study. The JECS was designed to follow children from the prenatal period to the age of 13 years. The participants were recruited between January 2011 and March 2014 from 15 regional centers throughout Japan, and the follow-up was mainly conducted via a self-administered questionnaire. The detailed protocol of the study and baseline profiles of the JECS participants have been reported previously[29,30]. The participants answered a questionnaire about lifestyle and behavior twice during pregnancy. The questionnaires completed at recruitment (early pregnancy) and later during mid- or late pregnancy were referred to as M-T1 and M-T2, respectively. The husbands of half of the participants also answered a self-administered questionnaire (F-T1) between the mothers' early pregnancy and 1 month after delivery. The F-T1 covered information such as lifestyle and medical history. The mean gestational weeks (standard deviation, SD) at the time of responding to M-T1 and M-T2 were 16.4 (8.0) and 27.9 (6.5) weeks, respectively. Participants also answered a questionnaire about their offspring 3 years after delivery (C-3y).

**Exposure 1: Maternal PA before and during pregnancy**. The Japanese short version of the international physical activity questionnaire (IPAQ), for which test-retest reliability and criterion validity were reported elsewhere, was used to evaluate maternal PA[31,32]. Participants reported their duration and frequency of PA lasting ≥10 min, divided by intensity into mild, moderate, and high. Based on that declaration, we estimated their mean PA per week before and during pregnancy in the M-T1 (based on recall) and M-T2 questionnaires, respectively. We calculated PA in terms of the metabolic equivalent of a task (MET), measured as the number of minutes per week (METs-min/week)[31]. PA, as defined in the IPAQ, includes all activities of daily life, such as work, housework, and leisure activities.

We divided the participants into five groups based on their prepregnancy PA levels. We also divided the participants into five groups based on their levels of PA during pregnancy. In each of the five groups, the PA = 0 group consisted of participants whose PA was 0. The other participants were divided into four groups using PA quartile points. The groups were labeled Quartiles 1–4 in ascending order of PA. Quartile 1 referred to the group with the lowest PA levels among the four groups, and Quartile 4 referred to the group with the highest PA levels. Further, the participants only reported PA that lasted ≥10 min; this implies that the PA = 0 group might not have reflected the actual PA. In addition, a general recommended amount of PA during pregnancy has not been determined. Therefore, in this study Quartile 1, which had the least amount of PA, was used as the control group instead of the PA = 0 group.

The abovementioned categorization of maternal PA was performed in the same way as in our previous study[24].

**Exposure 2: Maternal sleep before and during pregnancy**. In the M-T1 questionnaire, participants were asked about their

awakening time and bedtime before pregnancy. We calculated the sleep duration of participants and divided the participants into six groups according to sleep time: <6 h, 6–7 h, 7–8 h (reference), 8–9 h, 9–10 h, and >10 h. Participants were also divided into groups according to bedtime: 21:00 to 00:00 (reference), 24:00 to 03:00, and others (sleep before 21:00 or after 03:00). Since the bedtime for more than 95% of the analyzed participants was between 21:00 and 03:00 and the mode of bedtime was between 22:00 and 24:00, we further divided the participants into groups according to bedtime: 21:00, 24:00, and 03:00.

In the M-T2 questionnaire, participants were also asked about their usual awakening time and bedtime in the previous month. The participants were divided into groups as described above for M-T1.

The abovementioned categorization of maternal sleep was performed in the same way as in our previous research[25].

**Outcome: ASD in 3-year-old infants**. Three years after delivery, information about the infant was collected via the parent-reported questionnaire (C-3y). From among a list of diseases, the participants were asked to select those that their children had been diagnosed with by the age of 3 years. The list of diseases included ASD, which was expressed as "autistic spectrum disorder (e.g., autism, pervasive developmental disorder, Asperger's syndrome)." We defined outcome based on whether the participants selected ASD.

**Covariates**. Information on maternal age at delivery, prepregnancy body mass index (BMI), parity, gestational age at birth, infertility treatment, type of delivery, current history of HDP and diabetes/GDM, uterine infection, small for gestational age, and infant sex were collected from medical records transcripts. Parental information regarding smoking habits, alcohol consumption, educational background, history of psychiatric disorders (depression, anxiety disorders, and schizophrenia), autistic traits, and feeding status was collected via self-administered questionnaires.

The parents' autistic traits were evaluated by a short form of the Japanese version of the autism spectrum quotient (AQ-J-10)[33]. In concordance with previous studies, participants with a score of seven or more were categorized as having autistic traits.

**Statistics and reproducibility**. Between January 2011 and March 2014, 103,060 pregnancies were included in the study (Fig. 1). We limited the subject of this study to term-born singletons, since gestational age at birth and multiple fetuses are considered risk factors for autism[5,11]. The JECS study also includes several congenital anomalies. Since congenital abnormalities are considered to have a variable influence on infant development, cases with congenital anomalies were also excluded. We excluded 33,091 cases for the following reasons: previous participation in the study (n = 5647), multiple fetuses (n = 948), miscarriage or stillbirth (n = 3520), congenital anomaly or disease at 1 month of age (n = 3550), missing information on maternal age at delivery (n = 11), delivery before 37 weeks or after 42 weeks of gestation (n = 4454), and no response to all questions about maternal PA and sleep habits in the M-T1 and M-T2 (n = 751). Among the remaining 84,719 participants, 14,210 participants (17%) did not respond to the questionnaire regarding their 3-year-old offspring. Finally, the remaining 69,969 participants were included in the analysis. Among these, data of 37,145 fathers were available.

We used a log-binomial regression model to explore the association of each exposure with each outcome and to estimate the risk ratio (RR) of each outcome and the 95% confidence intervals (CIs). We initially adjusted for maternal age at delivery and then further adjusted for other maternal and child factors: smoking habits (never smokers, ex-smokers who quit before pregnancy, or smokers during early pregnancy), alcohol consumption (never drinkers, ex-drinkers who quit before pregnancy, or drinkers during early pregnancy), prepregnancy BMI (<18.5, 18.5–24.9, or ≥25.0 kg/m²), parity (0 or ≥1), infertility treatment (no ovulation stimulation/artificial insemination with husband's semen or assisted reproductive technology), maternal educational background (<10, 10–12, 13–16, or ≥17 years), maternal history of depression (yes or no), anxiety disorders (yes or no), and schizophrenia (yes or no), maternal autistic traits (yes or no), uterine infection (yes or no), type of delivery (vaginal or cesarean section), gestational age at birth (37, 38, 39, 40, or 41 weeks), small for gestational age (yes or no), infant sex (boy or girl), and feeding (breast milk, formula, or both). The covariates to be added to the multivariate model were determined by referring to previous literature on potential risk factors for ASD[5,11,34,35]. Furthermore, we did not complete the missing data. Thus, the multivariate analysis was limited to participants who had all the covariate data. However, the proportion of participants excluded from the multivariate analysis due to missing information about covariates was only 1%.

When significant associations were found, we also performed subgroup analyses that included the paternal factors (age, smoking, educational background, history of psychiatric disorders, and autistic traits) in the log-binominal model.

In this study, we used a fixed dataset "jecs-ta-20190930," which was released in October 2019. Stata version 16 (StataCorp LP, College Station, TX, USA) was used for all statistical analyses.

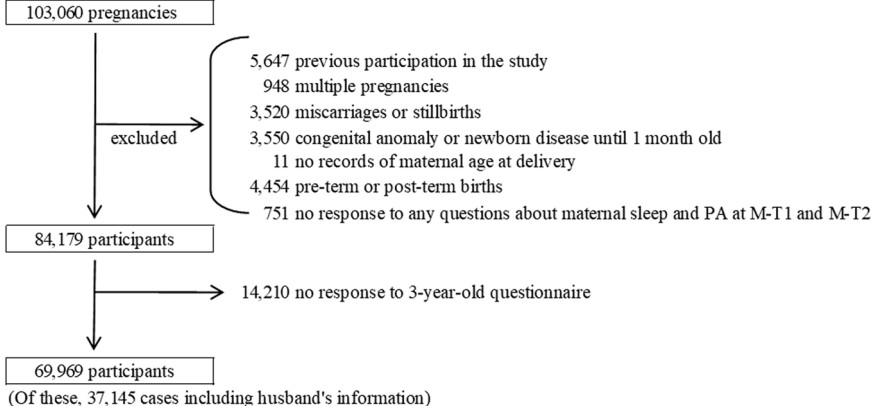

**Fig. 1 Study population flowchart.** PA physical activity, M-T1 questionnaire administered at recruitment, M-T2 questionnaire administered during mid- or late pregnancy.

**Table 1 Association between maternal physical activity before or during pregnancy and diagnosis of autism spectrum disorder in infants aged 3 years (parent-reported; Japan Environment and Children's Study).**

|  | No. of participants | No. of outcomes |  | Maternal age-adjusted model | | | Multivariable model[a] | | |
|---|---|---|---|---|---|---|---|---|---|
|  |  |  | % | RR | 95% CI | | RR | 95% CI | |
| Physical activity |  |  |  |  |  |  |  |  |  |
| Before pregnancy |  |  |  |  |  |  |  |  |  |
| 0 | 11,816 | 57 | 0.5 | 0.99 | 0.70 | 1.40 | 0.97 | 0.68 | 1.38 |
| Q1 | 14,144 | 70 | 0.5 | Reference | | | Reference | | |
| Q2 | 14,254 | 76 | 0.5 | 1.08 | 0.78 | 1.49 | 1.12 | 0.80 | 1.55 |
| Q3 | 14,096 | 58 | 0.4 | 0.85 | 0.60 | 1.20 | 0.85 | 0.60 | 1.21 |
| Q4 | 14,305 | 56 | 0.4 | 0.85 | 0.60 | 1.21 | 0.81 | 0.57 | 1.16 |
| During pregnancy |  |  |  |  |  |  |  |  |  |
| 0 | 15,210 | 80 | 0.5 | 0.94 | 0.68 | 1.29 | 0.91 | 0.66 | 1.26 |
| Q1 | 12,796 | 71 | 0.6 | Reference | | | Reference | | |
| Q2 | 12,281 | 48 | 0.4 | 0.71 | 0.49 | 1.02 | 0.69 | 0.48 | 0.99 |
| Q3 | 13,382 | 67 | 0.5 | 0.92 | 0.66 | 1.28 | 0.87 | 0.62 | 1.21 |
| Q4 | 12,839 | 43 | 0.3 | 0.63 | 0.43 | 0.92 | 0.61 | 0.42 | 0.90 |

*CI* confidence interval, *RR* risk ratio.
[a]Adjusted for maternal age at delivery, smoking habits, alcohol consumption, prepregnancy body mass index, parity, infertility treatment, maternal educational background, history of depression, anxiety disorders and schizophrenia, autistic traits, uterine infection, type of delivery, gestational age at birth, small for gestational age, infant sex, and feeding status.

**Reporting summary**. Further information on research design is available in the Nature Research Reporting Summary linked to this article.

## Results

Baseline characteristics of the participants along with available data on maternal PA and sleep duration during pregnancy are summarized in Supplementary Data 1. To determine the risk of potential bias due to missing data, we compared background characteristics between the analyzed population ($n = 69,969$) and the population excluded from the analysis due to non-response to the questionnaire after 3 years (C-3y) ($n = 14,210$) (Supplementary Data 2). The no-response group had more participants who were aged <25 years and had smoking habits and a lower educational background. However, the distributions of sleep duration and PA during pregnancy in the non-response group were similar to those in the analyzed population. Among the 69,969 participants analyzed, 322 (0.5%) were reported as having ASD.

**Association of maternal PA before and during pregnancy with ASD in 3-year-old infants**. The median (interquartile range) PA levels before and during pregnancy were 834 (198–2970) and 495 (66–1386) METs-min/week, respectively. The Spearman correlation coefficient for the raw data of PA before and during pregnancy was 0.46.

There was no significant association between maternal PA before pregnancy and infant ASD.

In contrast, the highest levels of PA (Quartile 4) during pregnancy were associated with a low RR for ASD compared with the reference group both in the model adjusted for only maternal age and in the multivariate model (RR = 0.61, 95% CI = 0.42–0.90 in the multivariate model; Table 1). In the subgroup analysis adjusting for paternal factors, the point estimates of RR did not change substantially (Supplementary Table 1).

**Association of maternal sleep before and during pregnancy with ASD in 3-year-old infants**. The reported sleep duration was on average between 7 and 8 h, both before and during pregnancy, and the bedtime of ~70% of the participants was between 21:00 and 24:00. When using the raw data, the Spearman correlation coefficient for the sleep duration before and during pregnancy was 0.58, and for one of the bedtimes was 0.72.

There was no significant association between maternal sleep duration before pregnancy and infant ASD.

In contrast, short sleep duration (<6 h) during pregnancy was associated with a higher RR for infant ASD compared with sleep duration of 7–8 h, both in the model adjusted for only maternal age and in the multivariate model (RR = 1.87, 95% CI = 1.21–2.90 in the multivariate model). Additionally, the infants of mothers who slept for >10 h during pregnancy had a higher RR for ASD compared with those of mothers who slept for 7–8 h in both the models (RR = 1.56, 95% CI = 1.00–2.48 in the multivariate model; Table 2). The subgroup analysis after adjusting for paternal factors did not change the association between maternal short (<6 h) and long sleep duration (<10 h) during pregnancy and infant ASD (Supplementary Table 1).

There was no significant association between maternal bedtime both before and during pregnancy and infant ASD.

## Discussion

This study investigated the associations of maternal PA and sleep before and during pregnancy with ASD in 3-year-old infants using data from a nationwide large-scale cohort study in Japan. This study suggested that higher maternal PA levels during pregnancy were associated with lower RRs for ASD in 3-year-old infants. Furthermore, the study also suggested that too short or too long maternal sleep durations during pregnancy were associated with higher RRs for ASD in 3-year-old infants. In contrast, prepregnancy maternal PA and sleep duration were not associated with infant ASD. These associations did not change in the subgroup analysis limited to the participants with information on paternal factors.

The greatest strength and novelty of this study is that, to the best of our knowledge, it is the first study to show a direct association between maternal PA and sleep duration during pregnancy and the diagnosis of ASD in children. In addition, paternal factors could also be adjusted in half of the analyzed population in this study.

Maternal short or long sleep duration during pregnancy increases perinatal complications, such as GDM and preterm birth[9,10,19], which are known risk factors for ASD[11]. Sleep disorders, such as sleep-disordered breathing, affect offspring development, as evidenced by disrupted social skills and low

**Table 2 Association between maternal sleep before or during pregnancy and diagnosis of autism spectrum disorder in infants aged 3 years (parent-reported; Japan Environment and Children's Study).**

| | | No. of participants | No. of outcomes | | Maternal age-adjusted model | | | Multivariable model[a] | | |
|---|---|---|---|---|---|---|---|---|---|---|
| | | | No. | % | RR | 95% CI | | RR | 95% CI | |
| Before pregnancy | Sleep duration (hours) | | | | | | | | | |
| | <6 | 4590 | 29 | 0.6 | 1.38 | 0.92 | 2.08 | 1.30 | 0.86 | 1.96 |
| | 6–7 | 13,809 | 73 | 0.5 | 1.15 | 0.85 | 1.54 | 1.09 | 0.80 | 1.47 |
| | 7–8 | 23,659 | 108 | 0.5 | Reference | | | Reference | | |
| | 8–9 | 17,124 | 63 | 0.4 | 0.81 | 0.60 | 1.11 | 0.91 | 0.66 | 1.25 |
| | 9–10 | 6966 | 29 | 0.4 | 0.94 | 0.62 | 1.42 | 1.13 | 0.75 | 1.72 |
| | >10 | 3018 | 17 | 0.6 | 1.38 | 0.83 | 2.31 | 1.44 | 0.84 | 2.41 |
| | Bedtime | | | | | | | | | |
| | 21:00–24:00 | 46,913 | 286 | 0.6 | Reference | | | Reference | | |
| | 24:00–03:00 | 20,431 | 133 | 0.7 | 1.27 | 1.01 | 1.61 | 1.08 | 0.85 | 1.39 |
| | Other | 1822 | 15 | 0.8 | 1.73 | 0.97 | 3.10 | 1.41 | 0.75 | 2.50 |
| During pregnancy | Sleep duration (hours) | | | | | | | | | |
| | <6 | 3267 | 26 | 0.8 | 2.04 | 1.32 | 3.16 | 1.87 | 1.21 | 2.90 |
| | 6–7 | 10,490 | 53 | 0.5 | 1.28 | 0.91 | 1.81 | 1.20 | 0.85 | 1.70 |
| | 7–8 | 21,737 | 85 | 0.4 | Reference | | | Reference | | |
| | 8–9 | 19,739 | 83 | 0.4 | 1.08 | 0.80 | 1.46 | 1.18 | 0.87 | 1.60 |
| | 9–10 | 9801 | 47 | 0.5 | 1.26 | 0.88 | 1.80 | 1.46 | 1.03 | 2.11 |
| | >10 | 4354 | 24 | 0.6 | 1.57 | 1.00 | 2.47 | 1.56 | 1.00 | 2.48 |
| | Bedtime | | | | | | | | | |
| | 21:00–24:00 | 51,213 | 216 | 0.4 | Reference | | | Reference | | |
| | 24:00–03:00 | 16,676 | 98 | 0.6 | 1.44 | 1.13 | 1.83 | 1.21 | 0.94 | 1.55 |
| | Other | 1499 | 4 | 0.3 | 0.67 | 0.25 | 1.81 | 0.64 | 0.24 | 1.71 |

CI confidence interval, RR risk ratio.
[a]Adjusted for maternal age at delivery, smoking habits, alcohol consumption, prepregnancy body mass index, parity, infertility treatment, maternal educational background, history of depression, anxiety disorders and schizophrenia, autistic traits, uterine infection, type of delivery, gestational age at birth, small for gestational age, infant sex, and feeding status.

reading-test scores[36,37]. Although these reports implied the association between maternal sleep duration during pregnancy and offspring development, no reports have demonstrated this association directly. On the contrary, high PA during pregnancy not only reduces perinatal complications but also has a positive effect on infant development[7,8]. However, there have been no reports showing a direct association between maternal PA during pregnancy and infant ASD.

Furthermore, the association of preconception maternal PA and sleep duration with ASD has not been previously reported. Previous studies have shown the associations of prepregnancy maternal PA and sleep with sleep or developmental problems in 1-year-old infants[24,25]. However, prepregnancy maternal PA and sleep duration were not associated with ASD in 3-year-old infants in this study. This suggests that PA and sleep duration may be more important during than before pregnancy. Because there are too few studies to conclude whether prepregnancy maternal lifestyle is associated with offspring developmental disorders, further studies are warranted.

There seem to be several possible mechanisms underlying the association of maternal PA and sleep duration during pregnancy with infant ASD. One of these mechanisms involve inflammation. Inflammation in the uterus is thought to be a cause of developmental disorders[16,17]. Exercise has an anti-inflammatory effect[13] and is reported to reduce inflammatory cytokines in pregnant women[12]. In contrast, too short or too long sleep durations increase inflammatory cytokine levels[14,15]. Both maternal PA and sleep seem to affect maternal inflammatory status and, thus fetal neurodevelopment.

Another mechanism to explain the results of this study is that maternal PA and sleep duration affect metabolism. Short sleep duration causes obesity and dyslipidemia, leading to hypertension and diabetes mellitus[15]. On the other hand, high levels of PA or exercise prevent the development of these diseases[18]. Similarly, in pregnant women, metabolic disorders, such as obesity and dyslipidemia, have been reported as risk factors for GDM and HDP[38]. In fact, short sleep duration increases the risk of developing GDM and HDP[19–21], and PA such as exercise prevents these complications[22,23]. Maternal obesity, GDM, and HDP have been reported as risk factors for ASD[11,39,40]. Maternal PA and sleep duration may affect the infant's neurodevelopment through maternal metabolism, blood pressure, and glucose tolerance.

This study also has some limitations. First, this was an observational study. Therefore, there could be unmeasured confounding factors, such as parental life rhythms and habits, especially after birth. Second, information on maternal lifestyle factors was collected using self-reported questionnaires. In particular, the information about prepregnancy maternal PA and sleep habits was reported at recruitment during pregnancy. PA was assessed using a validated international questionnaire, although the questions about maternal sleep have not been previously validated. These issues could have introduced measurement bias. For the outcome, we did not evaluate cases diagnosed after 3 years of age in this study. The reported cumulative percentage of patients with ASD diagnosed by the age of 3 is ~40% in Japan[41]. Therefore, the number of infants with ASD appeared to be underestimated in this study compared to those previously reported in Japan[41,42]. In addition, we were unable to confirm who made the ASD diagnoses and how they did so. For example, because the information for outcomes in this study was based on the answers of parents, the outcomes may have included children with positive results from ASD screening tests but without definitive diagnosis of ASD. The Modified Checklist for Autism in Toddlers is frequently used for ASD screening tests in early infancy, and its positive predictive value has been reported to be 0.455 in the Japanese population[43]. Therefore, the outcome data

in this study may be both underestimated and overestimated. As mentioned above, the inability to validate the ASD diagnosis in this study was also one of its significant limitations. Finally, ~17% of the participants were excluded from the analysis because they did not respond to the C-3y questionnaire, and this group of non-responders tended to be younger and comprise more smokers compared with the group of mothers who responded. Therefore, the group to be analyzed in this study might have had different characteristics compared with those of the population of the JECS study. This may have distorted the observed association between the maternal lifestyle during pregnancy and infant ASD.

In conclusion, higher maternal PA levels during pregnancy may be associated with a lower risk of ASD in 3-year-old infants, whereas too short or too long sleep durations during pregnancy may be associated with a higher risk of ASD in 3-year-old infants. An improvement in maternal lifestyle habits during pregnancy may have a positive impact on child development.

### Data availability
Data are unsuitable for public deposition due to ethical restrictions and legal framework of Japan. It is prohibited by the Act on the Protection of Personal Information (Act No. 57 of 30 May 2003, amendment on 9 September 2015) to publicly deposit the data containing personal information. Ethical Guidelines for Medical and Health Research Involving Human Subjects enforced by the Japan Ministry of Education, Culture, Sports, Science, and Technology and the Ministry of Health, Labour and Welfare also restricts the open sharing of the epidemiologic data. All inquiries about access to data should be sent to: jecs-en@nies.go.jp. The person responsible for handling enquiries sent to this e-mail address is Dr Shoji F. Nakayama, JECS Programme Office, National Institute for Environmental Studies.

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

## Acknowledgements

We would like to express our gratitude to all the participants of this study and all individuals involved in data collection. The Japan Environment and Children's Study was funded by the Ministry of Environment, Japan. The findings and conclusions of this article are solely the responsibility of the authors and do not represent the official views of the above government. This work was inspired by other works supported by the RIKEN Healthcare and Medical Data Platform Project and JSPS KAKENHI (grant numbers: JP16H01880, JP16K13072, JP18H00994, and JP18H03388).

## Author contributions

Study conception and design: S.M. Statistical analyses: T.M. Drafting of the manuscript and approval of final content: K.N., S.M., and T.M. Critical revision of the manuscript for important intellectual content and manuscript review: K.N., T.M., S.M., N.H., M.O., K.K. (Kiyoko Kato), M.S. (Masafumi Sanefuji), E.S., M.T., M.S. (Masayuki Shimono), T.K., S.O., K.K. (Koichi Kusuhara), and JECS group members.

## Competing interests

The authors declare no competing interests.

## Additional information

## the Japan Environment and Children's Study Group

Michihiro Kamijima[10], Shin Yamazaki[11], Yukihiro Ohya[12], Reiko Kishi[13], Nobuo Yaegashi[14], Koichi Hashimoto[15], Chisato Mori[16], Shuichi Ito[17], Zentaro Yamagata[18], Hidekuni Inadera[19], Takeo Nakayama[20], Hiroyasu Iso[21], Masayuki Shima[22], Youichi Kurozawa[23], Narufumi Suganuma[24] & Takahiko Katoh[25]

[10]Nagoya City University, Nagoya, Japan. [11]National Institute for Environmental Studies, Tsukuba, Japan. [12]National Center for Child Health and Development, Tokyo, Japan. [13]Hokkaido University, Sapporo, Japan. [14]Tohoku University, Sendai, Japan. [15]Fukushima Medical University, Fukushima, Japan. [16]Chiba University, Chiba, Japan. [17]Yokohama City University, Yokohama, Japan. [18]University of Yamanashi, Chuo, Japan. [19]University of Toyama, Toyama, Japan. [20]Kyoto University, Kyoto, Japan. [21]Osaka University, Suita, Japan. [22]Hyogo College of Medicine, Nishinomiya, Japan. [23]Tottori University, Yonago, Japan. [24]Kochi University, Nankoku, Japan. [25]Kumamoto University, Kumamoto, Japan.

