## [Peer Review File · Communications Medicine]

Referee expertise:

Referee #1: PhD, epidemiology, exercise

Referee #2: MD, PhD, ASD, childhood, psychiatry

Reviewers' comments:

Referee #1: Comments for Authors:

I enjoyed reading this piece. This is a large prospective study on the association between maternal physical activity/sleep duration and children's ASD. I have a few comments to help improve the already excellent article.

1. Differently from others, this is a topic for which randomized studies are doable. I suggest authors to say that up in front and mention that such studies are not available. Authors acknowledge the limitation of using observational data, which is adequate.
2. Causal mechanisms are only mentioned towards the middle of the discussion section. I suggest authors to bring a summary of possible causal pathways to the introduction section.
3. In the results, I think it is confusing to start saying there was not association between physical activity and ASD, and the following sentence will say there was as association between high physical activity and lower risk of ASD. What authors probably want to say with the first sentence is that there was no overall association between the two variables (using a linear trend of heterogeneity test?). It is important to clarify these issues both here and in the following paragraph (the same thing happens to sleep).
4. I suggest authors to explain the reasons for the exclusion criteria used in the project. For example, preterm infants were excluded, apparently. Why? It is not that I disagree with it, I just think readers should understand the reasons.
5. In the sleep analysis, it is justifiable to use a group in the middle as the reference category. In the physical activity one, I could not understand the rational for doing that. Although authors briefly try to provide a rationale for that decision, I would urge them to rethink the approach of the justification.
6. The lack of a hierarchical conceptual framework to guide the analysis is a limitation of the paper. Adjusted models include variables that are within the likely pathway from exposure to outcome, so that they should not be treated confounders. I suggest the authors to have a look at Prof. Victora and colleagues' piece in the Int J Epidemiol about the use of conceptual frameworks (published in 1997 or 1998).

Referee #2: Comments for Authors:

This study examines whether maternal physical activity and sleep before and during pregnancy are associated with infant ASD diagnosis at age 3 using the data from the Japan Environment and Children's Study (JECS). JECS included >100,000 pregnant women, and follow-up data have been collected using self-report. Authors reported that higher physical activity level during pregnancy

decreased risks for ASD whereas too short or long sleep durations during pregnancy increased risks for ASD in their offspring.

This is a study with impressive sample size and well-organized multisite research effort using prospective follow-up design. However, enthusiasm for the study findings is significantly dampened by several methodological concerns that need to be addressed by the authors.

1. Study population

a. It is unclear why multiple fetus and congenital anomaly/disease at 1 month of age were excluded from the study population.

b. While 17% attrition in younger, or smoker mothers is not insignificant, authors appropriately mentioned it in the section of study limitation.

2. Outcome: ASD diagnosis

a. ASD prevalence of 0.5 % in this study population is much lower than those reported in the previous ASD epidemiological studies in Japan (Honda et al, 2005; Saito et al, 2020).

b. While this low prevalence in current study population could be accurate, it is more likely that this significant discrepancy of ASD prevalence rose from two sources of biases – sampling bias and information bias on ASD diagnosis.

c. It is difficult to examine validity and reliability of the ASD diagnosis in the current study due to lack of information on how the self-report data on ASD diagnosis was collected.

3. Covariates: Information on psychiatric disorders

a. Authors described history of psychiatric disorders as a binary variable. It is unclear how this variable was defined and what psychiatric disorders were included in a questionnaire used.

b. Information on psychiatric disorders is important because it is directly related to physical activity, and sleep duration of the mother. Indeed, often changes in physical activity and sleep are symptoms of psychiatric disorders.

4. Predictors: Physical activities and sleep duration

a. Since physical activity and sleep during pregnancy are highly likely to be correlated with those before pregnancy, it is difficult to understand the current study findings. It would be very informative to examine whether these predictors were indeed highly correlated before and during pregnancy.

b. Given the study findings, it is possible that rather than absolute physical activity level and sleep duration, but the changes of the patterns in physical activities and sleep during pregnancy are associated with offspring ASD. Such changes could be markers of psychiatric disorders and/or other medical conditions during pregnancy.

5. These concerns raise doubts about the conclusions of the study findings, because observed association are potentially explained by unmeasured confounders including maternal psychiatric disorders and genetic risks.

Point-by-point responses to the comments

We are grateful to the editor and the reviewers for spending time in reviewing our manuscript and for their useful comments. The comments have been included below in *italics*, and our responses are in **bold font**. Changes made in the manuscript in response to the comments are in **red font**.

Referee #1: Comments for Authors:

I enjoyed reading this piece. This is a large prospective study on the association between maternal physical activity/sleep duration and children's ASD. I have a few comments to help improve the already excellent article.

(Response)

We really appreciate your effort regarding the review of our manuscript and your insightful comments.

1. Differently from others, this is a topic for which randomized studies are doable. I suggest authors to say that up in front and mention that such studies are not available. Authors acknowledge the limitation of using observational data, which is adequate.

(Response)

Thank you for your comment. In the Introduction, we have included information regarding the association between maternal physical activity or sleep and infant's ASD, which has not been reported previously in either interventional or observational studies.

(page 6, line 83–85)

2. Causal mechanisms are only mentioned towards the middle of the discussion section. I suggest authors to bring a summary of possible causal pathways to the introduction section.

(Response)

Thank you for your suggestion. We have added a summary of the possible causal pathways in the Introduction section.

(page 5, line 75–81)

3. In the results, I think it is confusing to start saying there was not association between physical activity and ASD, and the following sentence will say there was as association between high physical activity and lower risk of ASD. What authors probably want to

say with the first sentence is that there was no overall association between the two variables (using a linear trend of heterogeneity test?). It is important to clarify these issues both here and in the following paragraph (the same thing happens to sleep).

(Response)

We intended to state that pre-pregnancy PA and sleep were not associated with the outcomes, and that PA and sleep during pregnancy were associated with the outcomes.

However, as you have pointed out, the text in the Results section might have been confusing for the readers.

We have revised the term “pre-pregnancy” to “before pregnancy” in the Results section to improve clarity.

(pages 15–16)

4. I suggest authors to explain the reasons for the exclusion criteria used in the project. For example, preterm infants were excluded, apparently. Why? It is not that I disagree with it, I just think readers should understand the reasons.

(Response)

Thank you for your comment. We have clarified the reason for limiting this study to term-born singletons.

(page 8, line 124–128)

“We limited the subject of this study to term-born singletons, since gestational age at birth and multiple fetuses are considered risk factors for autism.^{5,11} The JECS study also includes several congenital anomalies. Since congenital abnormalities are considered to have a variable influence on infant development, cases with congenital anomalies were also excluded.”

5. In the sleep analysis, it is justifiable to use a group in the middle as the reference category. In the physical activity one, I could not understand the rationale for doing that. Although authors briefly try to provide a rationale for that decision, I would urge them to rethink the approach of the justification.

(Response)

In this study, the participants only reported physical activity lasting ≥ 10 minutes. Therefore, the group with PA = 0 might not have reflected the actual physical activity. In addition, a general recommended amount of physical activity for pregnant women has not been determined.

Therefore, the Quartile 1 group, which had the least amount of physical activity, was defined as the control group, instead of the “PA = 0” group.

We have added this in the Methods section.

(page 10, line 157–161)

6. The lack of a hierarchical conceptual framework to guide the analysis is a limitation of the paper. Adjusted models include variables that are within the likely pathway from exposure to outcome, so that they should not be treated confounders. I suggest the authors to have a look at Prof. Victora and colleagues' piece in the Int J Epidemiol about the use of conceptual frameworks (published in 1997 or 1998).

(Response)

Thank you for your suggestion and an introduction to the referenced literature. Although the etiology of ASD has not been elucidated, several factors have been reported to be associated with autism. Therefore, we considered that it would be difficult to create an accurate hierarchical framework. However, as you have pointed out, hypertensive disorders during pregnancy (HDP) and gestational diabetes mellitus (GDM) are clearly intermediate factors on the causal pathway. Accordingly, we excluded HDP and GDM as covariates in the multivariate analysis.

Due to this change, some values of RR and 95%CI in the Results section and the Table have been changed. We have also revised the text regarding HDP and GDM in the Discussion. (page 19, line 317–323)

Referee #2: Comments for Authors:

This study examines whether maternal physical activity and sleep before and during pregnancy are associated with infant ASD diagnosis at age 3 using the data from the Japan Environment and Children's Study (JECS). JECS included >100,000 pregnant women, and follow-up data have been collected using self-report. Authors reported that higher physical activity level during pregnancy decreased risks for ASD whereas too short or long sleep durations during pregnancy increased risks for ASD in their offspring.

This is a study with impressive sample size and well-organized multisite research effort using prospective follow-up design. However, enthusiasm for the study findings is significantly dampened by several methodological concerns that need to be addressed by the authors.

(Response)

Thank you for your meaningful comments. We have addressed your concerns in the responses below:

1. Study population

a. It is unclear why multiple fetus and congenital anomaly/disease at 1 month of age were excluded from the study population.

b. While 17% attrition in younger, or smoker mothers is not insignificant, authors appropriately mentioned it in the section of study limitation.

(Response)

Thank you for your comments. We have clarified the reason for limiting this study to term-born singletons in the Methods section.

(page 8 line 124–128)

“We limited the subject of this study to term-born singletons, since gestational age at birth and multiple fetuses are considered risk factors for autism.^{5,11} The JECS study also includes several congenital anomalies. Since congenital abnormalities are considered to have an influence on infant development, cases with congenital anomalies were also excluded.”

We have also added in the limitations that the group that was analyzed in this study might have had different characteristics compared with the population of the JECS study.

(page 20 line 342–344)

2. Outcome: ASD diagnosis

a. ASD prevalence of 0.5 % in this study population is much lower than those reported in the previous ASD epidemiological studies in Japan (Honda et al, 2005; Saito et al, 2020).

b. While this low prevalence in current study population could be accurate, it is more likely that this significant discrepancy of ASD prevalence rose from two sources of biases – sampling bias and information bias on ASD diagnosis.

c. It is difficult to examine validity and reliability of the ASD diagnosis in the current study due to lack of information on how the self-report data on ASD diagnosis was collected.

(Response)

Thank you for pointing this out to us.

The outcome of this study was diagnosis of ASD by the age of 3 years. Children who were diagnosed with ASD after 3 years were not evaluated. Therefore, the number of children with ASD appeared to be lesser than those previously reported in Japan.

Furthermore, as you have pointed out, we were unable to confirm who made the ASD diagnosis and how. This might have included information bias.

We have revised the limitations of this study in the Discussion to include the aforementioned points.

(page 20 line 331–335)

3. Covariates: Information on psychiatric disorders

a. Authors described history of psychiatric disorders as a binary variable. It is unclear how this variable was defined and what psychiatric disorders were included in a questionnaire used.

b. Information on psychiatric disorders is important because it is directly related to physical activity, and sleep duration of the mother. Indeed, often changes in physical activity and sleep are symptoms of psychiatric disorders.

(Response)

Thank you for your comments.

We have reclassified maternal psychiatric disorders into depression, anxiety disorder, and schizophrenia, which are reportedly risk factors for childhood ASD*.

The number of participants with schizophrenia was very few; therefore, we only adjusted maternal depression (yes or no) and anxiety disorders (yes or no) in the multivariate analysis.

We have added this in the Methods section and have shown the number of participants with psychiatric disorders for each disease in Table 1.

(page 13, line 206 and page 13, line 212–215 and Table 1)

4. Predictors: Physical activities and sleep duration

a. Since physical activity and sleep during pregnancy are highly likely to be correlated with those before pregnancy, it is difficult to understand the current study findings. It would be very informative to examine whether these predictors were indeed highly correlated before and during pregnancy.

b. Given the study findings, it is possible that rather than absolute physical activity level and sleep duration, but the changes of the patterns in physical activities and sleep

during pregnancy are associated with offspring ASD. Such changes could be markers of psychiatric disorders and/or other medical conditions during pregnancy.

(Response)

As per your suggestion, we have added the Spearman correlation coefficient for maternal PA and sleep habits before and during pregnancy. There appeared to be some correlation between maternal lifestyle before and during pregnancy.

(page 15 line 241–242 and line 254–255)

As you have pointed out, the influence of changes in the mother’s lifestyle is noteworthy. However, we think that such an analysis in this study might have been difficult for two reasons:

First, since there was some correlation between maternal PA or sleep habits before and during pregnancy, classification according to change in maternal lifestyle was difficult.

Second, while it is possible to limit the participants to a specific amount of physical activity or sleep duration before pregnancy and analyse the influence of change in maternal lifestyle, the results of that analysis could be difficult to apply to the whole study population due to the difference in population background.

5. These concerns raise doubts about the conclusions of the study findings, because observed association are potentially explained by unmeasured confounders including maternal psychiatric disorders and genetic risks.

(Response)

We have made relevant revisions in the manuscript to resolve the concerns of the reviewers.

However, since this study was an observational study, we think that the effects of unmeasured confounding factors and some bias due to the study design were completely unavoidable.

We hope that this revision will make it easier for the readers to understand the significance of this study, including its limitations.

Thank you.

Reviewers' comments:

Reviewer #1 (Remarks to the Author):

The authors have adequately addressed my comments

Reviewer #2 (Remarks to the Author):

Authors put efforts to address concerns this reviewer raised in prior review.
These are remaining concerns.

1. Study population

Authors commented that 17% attrition might have resulted in different characteristics than that of the J ECS study. More significant concern is that there is a possibility that this differential attrition in younger and smoker mothers might have distorted the observed association between PA and sleep during pregnancy and risks for ASD. This should be noted in the manuscript.

2. Outcome: ASD diagnosis

a. Authors contributed the discrepancy between ASD prevalence of 0.5 % in this study population and that were reported in the previous ASD epidemiological studies in Japan to younger age in study participants. Citation is needed to support it.

b. Inability to validate the ASD diagnosis in this study is a significant one. This needs to be clearly commented and how this might have affected observed association.

3. Covariates: Information on psychiatric disorders

a. I disagree with authors not to adjust maternal schizophrenia because of small number. 98 is not a small number, and furthermore, schizophrenia is a much stronger risk factor for ASD than anxiety or depression. This needs to be included in the psychiatric history of the mothers.

4. Predictors: Physical activities and sleep duration

a. Authors used spearman correlation between PA and sleeps. It is unclear if they used categorization of the variables or raw data for correlational analyses. If authors used categorized variables, it is unsurprising to see rather modest correlation coefficient. I would recommend rather to use raw data for Spearman's correlation if variables are not linear and Pearson correlation if they are linear, which will provide more accurate relationship.

b. The reasons for difficulties to use changes in the PA and/or sleep in the analyses are hard to understand. Why would correlation and different background of participants make examination of PA/sleep changes difficult?

5. While it is true that observational studies may not avoid unmeasured confounding factors including unmeasured genetic risks, it is still very important to comment those limitations of the study design so the readers could judge the findings of the study based with the information. Based on current inherent limitations of the observational study design along with specific method limitation if the current study, the conclusions of the study are not warranted and needs to be toned down to address them as associations, not causal mechanisms. For example, authors commented "higher PA levels during pregnancy may reduce ASD" in line 345-346. Current study does not support that higher PA levels during pregnancy reduce ASD risks, but it is associated with lower risk of ASD. This is a significant difference. Causal statement such as "reducing risk for ASD" is misleading.

Point-by-point responses to the comments

We are grateful to the editor and reviewers for spending time reviewing our manuscript and for their useful comments. The comments have been included below in *italics*, and our responses are in **bold font**. Changes made to the manuscript in response to the comments are in **red font**.

Referee #1 (Remarks to the Author):

The authors have adequately addressed my comments.

Referee #2 (Remarks to the Author):

Authors put efforts to address concerns this reviewer raised in prior review. These are remaining concerns.

1. Study population

Authors commented that 17% attrition might have resulted in different characteristics than that of the JECS study. More significant concern is that there is a possibility that this differential attrition in younger and smoker mothers might have distorted the observed association between PA and sleep during pregnancy and risks for ASD. This should be noted in the manuscript.

(Response)

Thank you for your suggestion.

We posit that it is not possible to determine whether differences in group characteristics would strengthen or weaken the associations between maternal lifestyle during pregnancy and the risks for ASD, because maternal advanced age and smoking were reported as possible risk factors for ASD.

As you suggested, we have simply noted the possibility for distortion of the results in the discussion section of the manuscript and would like to leave the interpretation up to the reader.

“Therefore, the group to be analyzed in this study might have had different characteristics compared with those of the population of the JECS study. This might have ~~introduced some selection bias~~ distorted the observed association between the maternal lifestyle during pregnancy and infant ASD.” (line 342 -345)

2. Outcome: ASD diagnosis

a. Authors contributed the discrepancy between ASD prevalence of 0.5 % in this study population and that were reported in the previous ASD epidemiological studies in Japan to younger age in study participants. Citation is needed to support it.

(Response)

Thank you for your comment.

According to the previous study in Japan, the cumulative percentage of ASD patients diagnosed by the age of 3 was about 40%. We have added the study as a citation.

“For the outcome, we did not evaluate the cases diagnosed after 3 years of age in this study; however, the reported cumulative percentage of ASD patients diagnosed by the age of 3 is approximately 40% in Japan¹. Therefore, the number of infants with ASD appeared to be underestimated in this study compared to those previously reported in Japan.” (line 325-329)

b. Inability to validate the ASD diagnosis in this study is a significant one. This needs to be clearly commented and how this might have affected observed association.

(Response)

We agree with your opinion.

Thus we rewrote the Discussion to explain that the outcome data in this study might be underestimated and overestimated due to inability to validate the ASD diagnosis.

“In addition, we were unable to confirm who made the ASD diagnoses and how they did so. For example, because the information for outcomes in this study was based on the answers of parents, the outcomes may have included children with positive results from ASD screening tests but without definitive diagnosis of ASD. The Modified Checklist for Autism in Toddlers (M-CHAT) is frequently used for ASD screening tests in early infancy, and its positive predictive value has been reported to be 0.455 in the Japanese population.² Therefore, the outcome data in this study may be both underestimated and overestimated. As mentioned above, the inability to validate the ASD diagnosis in this study was also one of its significant limitations. ~~Therefore, the outcome of this study may include misclassification.~~” (line 330 – 338)

3. Covariates: Information on psychiatric disorders

a. I disagree with authors not to adjust maternal schizophrenia because of small number. 98 is not a small number, and furthermore, schizophrenia is a much stronger

risk factor for ASD than anxiety or depression. This needs to be included in the psychiatric history of the mothers.

(Response)

Thank you for your comment.

We have added “schizophrenia” as a covariate in the multivariate analysis. Some of the RRs and 95% CIs changed following this inclusion; however, our results and conclusion remained unchanged.

(line 244 and line 258-261)

4. Predictors: Physical activities and sleep duration

a. Authors used spearman correlation between PA and sleeps. It is unclear if they used categorization of the variables or raw data for correlational analyses. If authors used categorized variables, it is unsurprising to see rather modest correlation coefficient. I would recommend rather to use raw data for Spearman’s correlation if variables are not linear and Pearson correlation if they are linear, which will provide more accurate relationship.

(Response)

Thank you for your suggestions. We have calculated Spearman’s correlation of based on the raw data of PA, sleep duration, and bedtime and revised the corresponding text accordingly (line 239 and line 251).

b. The reasons for difficulties to use changes in the PA and/or sleep in the analyses are hard to understand. Why would correlation and different background of participants make examination of PA/sleep changes difficult?

(Response)

Thank you for your comments. Our response to your previous comments regarding change of maternal lifestyle might have been confusing due to lack of a clear explanation.

It is methodologically possible to investigate associations between change of maternal PA or sleep duration (increase, not so change, decrease) and infant ASD. However, the effects of lifestyle changes may vary depending on pre-pregnancy PA and sleep time. For example, the effects of increased sleep duration may differ between participants who had a short or long sleep duration before pregnancy. PA

could be considered similarly. In addition, PA during pregnancy was generally lower than PA before pregnancy, and it was difficult to classify the study population in terms of the changes in PA amount.

To solve the abovementioned problem, a sub-group analysis to limit the participants with a specific amount of PA or sleep duration might be useful. One example is the analysis to limit the participants with adequate sleep duration (7-9h) before pregnancy. However, the results of sub-group analysis cannot be applied for all study population since the population backgrounds may be different between population of sub-group analysis and whole population in this study. Although it is methodologically possible to analyse by maternal lifestyle changes before and during pregnancy, we consider it to be difficult to interpret the result of that analysis.

5. While it is true that observational studies may not avoid unmeasured confounding factors including unmeasured genetic risks, it is still very important to comment those limitations of the study design so the readers could judge the findings of the study based with the information. Based on current inherent limitations of the observational study design along with specific method limitation if the current study, the conclusions of the study are not warranted and needs to be toned down to address them as associations, not causal mechanisms. For example, authors commented “higher PA levels during pregnancy may reduce ASD” in line 345-346. Current study does not support that higher PA levels during pregnancy reduce ASD risks, but it is associated with lower risk of ASD. This is a significant difference. Causal statement such as “reducing risk for ASD” is misleading.

(Response)

We agree that this study showed only associations and not causal mechanisms. However, as you have pointed out, there were some phrases in the manuscript that suggested causal relationships. We have revised such phrases.

“Higher levels of maternal physical activity and adequate sleep duration during pregnancy may ~~reduce~~ be associated with ASD in infants.” (Line 59- 60)

“In conclusion, higher maternal PA levels during pregnancy may ~~reduce~~ be associated with a lower risk of ASD in 3-year-old infants, whereas too short or too long sleep durations during pregnancy may ~~increase~~ be associated with a higher risk of ASD in 3-year-old infants.” (line 346-349)

REVIEWERS' COMMENTS:

Reviewer #2 (Remarks to the Author):

Authors addressed this reviewer's comments appropriately.

Point-by-point responses to the comments

We are grateful to the editor and reviewers for the time and effort involved in reviewing our manuscript. The comments have been included below in *italics*, and our responses are in **bold font**.

In this revision, there are no changes to the manuscript.

Referee #2 (Remarks to the Author):

Authors addressed this reviewer's comments appropriately.

(Response)

Thank you for the thorough review of our manuscript.